# Significant Carrier Extraction Enhancement at the Interface of an InN/p-GaN Heterojunction under Reverse Bias Voltage

**DOI:** 10.3390/nano8121039

**Published:** 2018-12-12

**Authors:** Vladimir Svrcek, Marek Kolenda, Arunas Kadys, Ignas Reklaitis, Darius Dobrovolskas, Tadas Malinauskas, Mickael Lozach, Davide Mariotti, Martin Strassburg, Roland Tomašiūnas

**Affiliations:** 1Research Center for Photovoltaics, National Institute of Advanced Industrial Science and Technology (AIST), Tsukuba 305-8568, Japan; mickael.lozach@aist.go.jp; 2Institute of Photonics and Nanotechnology, Faculty of Physics, Vilnius University, Sauletekio 3, 10257 Vilnius, Lithuania; mkolendaus@gmail.com (M.K.); arunas.kadys@ff.vu.lt (A.K.); ignas.reklaitis@gmail.com (I.R.); darius.dobrovolskas@gmail.com (D.D.); tadas.malinauskas@ff.vu.lt (T.M.); rolandas.tomasiunas@ff.vu.lt (R.T.); 3Nanotechnology & Integrated Bio-Engineering Centre (NIBEC), Ulster University, Shore Road, Newtownabbey BT37 0QB, UK; d.mariotti@ulster.ac.uk; 4OSRAM Opto Semiconductors GmbH, Leibnizstr. 4, D-93055 Regensburg, Germany; Martin.Strassburg@osram-os.com

**Keywords:** InN/p-GaN heterojunction, interface, photovoltaics

## Abstract

In this paper, a superior-quality InN/p-GaN interface grown using pulsed metalorganic vapor-phase epitaxy (MOVPE) is demonstrated. The InN/p-GaN heterojunction interface based on high-quality InN (electron concentration 5.19 × 10^18^ cm^−3^ and mobility 980 cm^2^/(V s)) showed good rectifying behavior. The heterojunction depletion region width was estimated to be 22.8 nm and showed the ability for charge carrier extraction without external electrical field (unbiased). Under reverse bias, the external quantum efficiency (EQE) in the blue spectral region (300–550 nm) can be enhanced significantly and exceeds unity. Avalanche and carrier multiplication phenomena were used to interpret the exclusive photoelectric features of the InN/p-GaN heterojunction behavior.

## 1. Introduction

Single-nitride solar cells possess key and important features that can facilitate their large-scale application, whereby the nontoxicity of the primary elements offers opportunities towards future “green” technologies. Of the nitride family, an attractive candidate is indium nitride (InN), which presents a narrow-energy band gap (*E_g_*) of 0.7 eV at room temperature, useful for the near-infrared region (NIR). Indium is also a key element in indium gallium nitride (InGaN) alloys, providing a bandgap for a broad spectral response from 0.7 eV to 3.4 eV without deteriorating its high carrier mobility [1,2]. The constituents of InN present a large difference in mass. Namely, the light nitrogen anions beside the heavy indium cations result in a phonon dispersion between high-lying optical phonon energies and low-lying acoustic phonon energies. Such a large gap [3] can effectively block Klemens decay [4,5]. Consequently, InN has been recognized as one of the most suitable materials with high potential for the realization of so-called hot carrier absorbers related to third-generation solar cell concept [6]. In principle, hot carrier absorbers are expected to reduce significant thermal losses by directly extracting the hot carriers [6,7]. Theoretically, their population is maintained inside the absorber by inhibiting the inherently ultrafast cooling process [6].

However, the realization of high-quality epitaxial InN layers remains a challenge. High-quality growth means very low dislocation density, and reduction of the unintended n-type doping due to indium or nitrogen vacancies and oxygen-related defects [8]. To date, the best results were obtained through epitaxial growth of InN on p-GaN by molecular beam epitaxy (MBE) [9]. From the point of view of optoelectronic device production, metalorganic vapor-phase epitaxy (MOVPE) is more suitable [10,11,12]. Nevertheless, MOVPE is restricted by the low decomposition rate of NH_3_, resulting in InN island growth on GaN, which deteriorates the quality of the InN films [13]. These nucleation islands also strongly affect the junction interface morphology by producing dislocation defects and causing inefficient photo-response of the devices (e.g., detectors, solar cells) [12].

In addition, the growth of high-quality GaN is favorable for fabrication of GaN avalanche photodiodes for optical detection in the ultraviolet spectral region, due to a low operation voltage and the possibility of Geiger-mode operation [14,15]. The combination of GaN with InN can expand the detection spectral range and enhance the avalanche properties. Recently, illuminated and biased GaN p-i-n avalanche photodiodes have been widely investigated to achieve low multiplication noise and high gain [16,17,18,19]. Under reverse bias, the extended depletion region generates a strong electric field, which effectively separates the electron–hole pairs. Although GaN avalanche photodiode properties have been widely reported [14,15,16,17,18,19], to our knowledge an InN/p-GaN junction has never been investigated in an avalanche regime. It should be stressed that only electron–hole pairs that are generated at the interface or very close to a high-quality InN/p-GaN junction can contribute to the external current.

The large lattice mismatch between GaN and InN hinders the growth of high-quality interfaces and leads to the formation of a diffusive interface with poor photoluminescence (PL)/electroluminescence (EL) emission and carrier photogeneration efficiency [20,21,22,23]. A diffusive interface is often attributed to the migration of indium and gallium atoms through the spinodal decomposition process resulting in the formation of clusters and localization of charge carriers. [24,25]. The intrinsic phonon properties of high-quality InN are thus of relevance to self-heating and phonon engineering. The phonon decay into two or more phonons is subject to energy and wave vector conservation, while basically two phonon decay phenomena are referred to as Klemens and Ridley phonon decay mechanisms [4,26]. The most probable decay channels are those for which the vibrations created have a high density of states, whereby the prevention of the Klemens optical phonon decay mechanism occurs at the mini-Brillouin-zone boundaries of nanostructured interfaces. The important feature in this context is a poor transmission of phonons across the interface. The development of crystal growth technology, resulting in persistent improvements in the crystal and interface quality, thus permits the examination of intrinsic properties and refinement of the important phonon decay mechanisms. In this context, the InN/GaN interface quality is a key factor for the development of the new generation of InN-based optoelectronic devices.

In this contribution, we demonstrate high-quality InN films with low defect concentration and limited-roughness interface with magnesium-doped p-GaN, both grown by pulsed MOVPE. The formed InN/p-GaN heterojunction shows good rectifying behavior with InN characteristic electron concentration, among the best values obtained by pulsed MOVPE. An efficient photoelectric InN/p-GaN heterojunction is demonstrated by measuring carrier extraction under unbiased and reverse biased conditions. 

## 2. Materials and Methods

The InN/p-GaN heterojunction investigated in this study was grown on a 5-μm-thick undoped GaN buffer layer and sapphire substrate by MOVPE (AIXTRON 3 × 2 CCS Flip Top reactor, AIXTRON SE, Herzogenrath, Germany). Trimethylgallium (TMGa), trimethylindium (TMIn) and ammonia (NH_3_) have been used as precursors for gallium (Ga), indium (In), and nitrogen (N), respectively. Bis(cyclopentadienyl) magnesium (Cp_2_Mg) was used as a precursor for Mg p-type doping. The GaN templates on sapphire were grown using a standard low-temperature GaN buffer layer followed by high-temperature GaN growth (i.e., two-step growth), which were the necessary processes to achieve a high-quality GaN film with low dislocation density and high carrier mobility (600 cm^2^/(V s) at 300 K) [27]. Then, a 600 nm p-GaN layer was grown on the un-doped 5-μm-thick GaN layer followed by 20 min annealing at 850 °C under a nitrogen (N_2_) ambient atmosphere to activate Mg acceptor states in the p-GaN. Finally, a 300-nm-thick InN layer was grown on the p-GaN under 400 mbar pressure using TMIn precursor using a multiple flow-interruption technique, that is, pulsed. One growth cycle duration was 27 s, where TMIn and NH_3_ were supplied into the reactor for the first 7 s and then only NH_3_ was supplied for the next 20 s. The overall number of growth cycles was 600. The growth process of the InN layers was split into two steps. InN began to grow at 570 °C for the first few dozen growth cycles; then the growth temperature was increased at a constant rate until 610 °C (first step). For the remaining growth cycles, the temperature was kept constant at 610 °C (second step). The V/III ratio was stabilized at 99/68 during the entire InN growth process.

Scanning electron microscopy (SEM; Apollo 300, CamScan, Cambridge, UK (now successor Applied Beams, LLC, Beaverton, OR, USA)) was used for evaluating the heterojunction cross-section. Electron beam induced current (EBIC; Digiscan II priedu, model 778, Gatan, Inc., Pleasanton, CA, USA) was used to determine the electrically active areas of the heterojunction cross-section.

In order to determine the crystallinity of the structure, X-ray diffraction (XRD) measurements were carried out using a Rigaku SmartLab X-ray diffractometer (Rigaku, Tokyo, Japan).

The optical properties were investigated by photoluminescence (PL) at room temperature using a continuous wave (cw) He-Ne laser (633 nm) as an excitation source. The PL emission was collected into a 0.3 m spectrometer (Andor, Shamrock 303, Oxford Instruments, Abingdon, UK) and detected by an InGaAs detector array (Andor, iDus DU491A-2.2, Oxford Instruments, Abingdon, UK. The Hall measurements were performed on the Van der Pauw Ecopia HMS-3000 Hall Measurement System (Bridge Technology, Chandler Heights, AZ, USA).

AM 1.5G standard solar spectra were simulated using Wacom Electric Co. solar simulator (JIS, IEC standard conforming, CLASS AAA, Tokyo, Japan) calibrated to give 100 mW/cm^2^ using two reference solar cells: a-Si and c-Si. The electrical data were recorded using a Keithley 2400 source meter (Tektronics, IL USA). The external quantum efficiency (EQE) characteristics were measured by CEP-25BXS (Bunkoh-Keiki Co., Ltd., Tokyo, Japan) in an extended spectral region of 300–2000 nm. 

## 3. Results and Discussion

The X-ray diffraction (XRD) 2-theta scan measurements of the InN/p-GaN structure (Figure 1a) revealed a pronounced peak at 31.46° corresponding to the diffraction of hexagonal InN (0002); compared to the relaxed InN (0002) [27,28,29] (vertical line at 31.33° in Figure 1a), our measurements show our InN layer slightly strained. The peak at 34.5° corresponds to hexagonal GaN (0002). Improved crystallinity is due to the pulsed nature of the growth process, which enabled higher temperature growth. This was confirmed by the rocking curve for the InN (0002) reflection (Figure 1b) with a full width at half maximum (FWHM) of ~0.284° (1022 arcs). This value is by far better than those reported in the literature for MOVPE growth: 1800 arcs (at 550 °C) [30], 5601 arcs (at 550 °C) [14], 1300 arcs [31] and 0.27° (at 550 °C) obtained for c-oriented prismatic InN nanowalls grown on c-GaN/sapphire [32]. The dislocation density estimated from the rocking curve FWHM [33] showed a total (screw and edge) value of ~6 × 10^10^ cm^−3^.

Figure 2a shows the structure of the investigated p-GaN/InN heterojunction and Figure 2b reports the SEM image of the junction edge. An electron-beam-induced current (EBIC) image is used to show the electrically active areas of the junction, that is, the depletion region with a built-in electric field. The non-equilibrium charge carriers (electrons, holes) in the depletion region, due to the built-in electric field, are immediately separated providing electrical current in the external circuit, expressed via bright spots on the image. Areas without an electric field remain dark (no electrical current), indicating fast recombination of the electron–hole pair as a consequence. In Figure 2c, we superimposed the EBIC on the SEM image to underline the precise location of the electrically active area, in our case, the junction between InN and p-GaN. The depletion region is quite shallow (<100 nm) and it follows the surface topology. 

Furthermore, the extent of the depletion region at the InN/p-GaN interface has been investigated theoretically. The built-in electrical field across the p–n interface is enhanced due to the gradient of spontaneous polarization. Figure 3a shows hole and electron concentration at the InN/p-GaN interface, and Figure 3b reports the built-in electrical field as a function of depth at the InN/p-GaN interface. The initial conditions for the simulations and doping concentration were evaluated from the equilibrium charge carrier concentrations obtained by Hall measurements. From our simulations, we estimate the thickness of the depletion region to be about 22.8 nm. The distribution of carrier concentration in the depletion region is broader towards p-GaN (18.3 nm) than InN (4.5 nm). We must underline that still the good quality of the InN film presented an unintended n-doping; to our knowledge, the measured electron concentration *n*_e_ = 5.19 × 10^18^ cm^−3^ and mobility *µ*_e_ = 980 cm^2^ V^−1^ s^−1^ are among the best values obtained by MOVPE growth (*n*_e_ = (3 ÷ 5) × 10^18^ cm^−3^, *µ*_e_ = (542 ÷ 980) cm^2^ V^−1^ s^−1^ [34,35,36]). 

The pulsed MOVPE growth method is quite commonly used to grow group III-nitride heterostructures [37,38,39], but in this case it allowed us to grow good crystalline-quality InN layers using a higher growth temperature (610 °C) (see Materials and Methods paragraph). The growth temperature in the MOVPE growth of InN is the most critical parameter to control the film quality. Because of the low InN dissociation temperature and high equilibrium nitrogen (N_2_) vapor pressure over the InN film, the preparation of InN requires a low growth temperature. Due to the low (400–500 °C) growth temperatures, the growth of InN is restricted by a low decomposition rate of NH_3_ and reduced migration of adatoms on the surface, which leads to metallic In formation on the surface [40,41]. In our sample, no metallic In droplets were observed on the surface of the InN layer. Due to higher growth temperature, the NH_3_ dissociation rate was enhanced and there was no shortage in active nitrogen atoms (N). The object input of high V/III ratio is to provide a sufficient amount of reactive nitrogen. The TMIn pulse length controls the thickness of the deposited ultrathin InN layer, while the pause length controls the time allowed for surface migration of In adatoms on the surface and the amount of additional reactive nitrogen. The ratio between the pause and pulse determines the effective V/III ratio, which can be expressed as $\frac{V}{III}={\left(\frac{V}{III}\right)}_{nom}\left(1+\frac{t_{p}}{t_{TMIn}}\right)$ [41]. Here, (*V/III*)*_nom_* is the nominal V/III ratio, in our case ~38,000 during the TMIn pulse of 7 s, *t_TMIn_* and *t_P_* are the TMIn pulse and pause lengths, respectively. A high V/III ratio and a high growth temperature for InN allowed us to achieve good electrical properties, such as low electron concentration and high mobility. This could be explained by the reduction in the native defect concentration, such as N vacancies [42]. In addition, the gradual increase of growth temperature (i.e., the temperature ramp) results in the repeated deposition of ultrathin InN layers, where each successive layer is deposited at a slightly higher temperature until the temperature reaches 610 °C. Such temperature ramping facilitates the formation of larger islands with better alignment due to the increased diffusion of In adatoms, determined by the ultrathin InN layers deposited at the lowest growth temperatures [41]. The temperature ramp approach is similar to using two-step growth with a low-temperature InN buffer [43] or a graded composition InGaN buffer [9], which are reported to improve the structural quality in the InN epilayers.

Furthermore, photoluminescence (PL) measurements of the structure at room temperature were performed. The PL spectra consist of two parts corresponding to the near-infrared (InN) (Figure 4a) and to the UV (p-GaN) (Figure 4b) spectral regions. The InN PL spectrum is centered at 1580 nm (0.75 eV) which is consistent with the established band-to-band PL of InN [44]. The PL spectrum of the p-GaN is a typical GaN room-temperature PL spectrum with near band-edge emission around 3.4 eV (360 nm) [45]. 

In order to study the capability of the heterojunction to generate a photo-current, two different types of measurements were performed: (i) photovoltaic response and (ii) EQE measurements. Figure 5a reveals the current density–voltage (J–V) characteristic of the InN/p-GaN heterojunction under dark and AM1.5G illumination conditions. The current density was measured for the applied bias voltage from −1 V to +1 V. The characteristics are asymmetrical with clear diode-like rectification behavior. Within the region of the turn-on voltage and when the voltage is above ~0.2 V under dark condition and above 0.06 V under illumination, the J–V characteristics follow a power law V ~ I^2^; consequently, for single-type carrier injection, the current conduction is expected to be space-charge limited. At larger applied voltages >0.5 V, the J–V does not deviate considerably from linearity, indicating low series resistance and retaining a low density of interface states. Figure 5b shows corresponding EQE spectra of the unbiased InN/p-GaN heterojunction. The carrier collection peaked in the wavelength region of 400–500 nm.

Figure 6a shows the EQE spectra of the InN/p-GaN junction at different reverse biases: 0 V, −10 V and −20 V. Photo-response in the region 300–600 nm originates from the absorption in both p-GaN and InN, as explained below. Local EQE maxima are peaked at wavelengths of 430 nm, 450 nm and 511 nm. Increased reverse bias results in a general rising trend of the full spectrum where the peak at 511 nm (2.4 eV) appears to be the strongest. The sub-maximum at around 360 nm originates from the bandgap transitions and is peaked at the same wavelength of the PL emission in Figure 4b; this implies that conditions are favorable for both charge carriers generated in the p-GaN to reach the contacts, that is, electrons down the barrier towards the n-contact and holes to the p-contact (see Figure 7a). The same applies to the broad band at 375–550 nm (3.3–2.2 eV), which correlates with known multiple transition mechanisms: (i) in the UV range peaked at 3.20–3.26 eV, the Mg doping introduces shallow acceptors with an ionization energy of 200 meV; (ii) in the blue spectra range peaked at 2.7–2.9 eV, the compensating deep donors formed at high p-doping levels are responsible for the donor-to-band or donor-acceptor-pair transitions [45]. The dissociation of electron–hole pairs enhanced under reverse bias is likely responsible for that. Surprisingly, the EQE in the 300–600 nm range is enhanced significantly above 100%. At −20 V reverse bias we could observe also a contribution to the EQE for the longer wavelengths (>1500 nm) (Figure 6b). The band peak position for carrier extraction found at 1780 nm (~0.7 eV) corresponded to the InN energy bandgap [44]. Figure 6c highlights the nonlinear increase of the photocurrent as a function of reverse bias. This suggests that the p–n junction is normally operated and the generated photocurrent increases exponentially with the applied bias attaining a 2kT/q slope in the semi-log (I–V) plot due to recombination effects [46].

In order to analyse the junction existing at the interface of InN/p-GaN, we have constructed a band diagram based on our measurement results and calculations (Figure 7a). The schematic energy band diagram of the InN/p-GaN heterojunction under irradiation is shown in Figure 7a, which reflects the type-I straddling configuration [47]. Nitride semiconductors are pyroelectrics, that is, they present a strong polarization due to fixed charges in the crystal structure [48,49]. The band diagram in Figure 7a shows the band bending behavior at the InN and p-GaN interface that underlines the heterojunction formation. The absorption of photons leads to the formation of excess electrons in the n-side and excess holes in the p-side of the device, generating a voltage drop Voc across the junction. The broader depletion region in the p-GaN maintains space charge neutrality, which results in efficient collection of the carriers generated by UV and blue–visible light (Figure 6). For electrons there is no offset, while for holes, the potential barrier of 0.3 eV is obtained (Figure 7a). Nevertheless, with 1600–1800 nm wavelengths, by 1 sun irradiation, electrons and holes are generated in the InN, and when a sufficient reverse bias (−20 V) is applied, holes can overcome the interface barrier and generate photocurrent for the near-infrared (NIR) region (Figure 7b). In the reverse bias, the presence of recombination centers affects the overall I–V characteristics by the generation of additional electron–hole pairs within the depletion region, which greatly exceeds the unbiased state. The generation rate of carriers in the depletion region will reach a maximum when E_g_/2. Since the intrinsic concentration is a function of the energy gap of the material, InN with a small energy gap will therefore exhibit high generation rates [50].

Although enhancement of the EQE over unity under different excitation conditions has been widely reported in the literature [51,52], the origin of the increase up to 1400% in EQE in InN/p-GaN heterojunction under −20 V reverse bias is not yet clear and may be attributed to various factors. The enhancement might originate from an enlargement of the depletion region contributing to the enhanced absorption and charge collection efficiency at the junction. Bias voltage actually increases the ionized donor concentration of GaN and decreases its work function while harvesting photo carriers in a globally larger volume for the light in the spectral region 300–600 nm. Likewise for carrier multiplication (CM), a larger depletion region ensures that any photogenerated carriers are efficiently collected [6,7,53]. 

The key aspect is to extract multiple excitons and hot carriers before they thermalize to the band edge and rapidly recombine. In principle, that could be possible in our case when the relative high-voltage reverse bias is applied. For the hot carrier contribution, however, InN should be most likely in a cubic structure featuring a very large gap between high-lying optical phonon and low-lying acoustic phonon energies. If the gap is wide enough, the Klemens decay of optical phonons is prevented [4,54]. However, a pronounced XRD peak at 31.3° for the InN film reveals the hexagonal lattice type, likewise for the GaN (0002) peak at 34.5° (Figure 1), weakening our expectation for the hot carrier contribution.

Our results show that the photocurrent is increasing nonlinearly with the reverse bias (Figure 6c) exhibiting a peak at 2.4 eV (see Figure 6b) which corresponds roughly to three times the InN bandgap. This observation might support further the contribution from CM, which was demonstrated in bulk InN to occur at photon energies roughly up to three times its bandgap [53,55]. When the reverse bias approaches the breakdown level, amplification factors at the electrical level can take place [56]. In order to get a better picture about these processes, we have simulated InN/p-GaN heterojunction at different reverse biases (Figure 7b,c). Because the depletion layer is relatively thin (in the range of nanometers), electrons can directly tunnel across the depletion region from p-GaN valence band (*E_V_*) into the n-InN conduction band. Due to the increased reverse bias (>−10 V), photocurrent generation proceeds at the breakdown level leading to the separation of multiple excitons and efficient electron–hole generation, therefore contributing significantly to the EQE of the heterojunction. Because of the large conduction band offset of the nitride materials, the high level of Mg doping improves the tunneling and therefore overall photocurrent generation at the InN/p-GaN heterojunction [57].

## 4. Conclusions

In summary, efficient electron–hole pair generation and extraction in a superior-quality InN/p-GaN heterojunction interface grown by pulsed MOVPE was achieved. The considerably improved crystallinity was obtained thanks to pulsed nature and higher-temperature growth process. The InN/p-GaN heterojunction depletion region was simulated, demonstrating 22.8 nm width extended by four times more into p-GaN than InN. Experimentally, good rectifying behavior and photo-carrier extraction without external electrical field was observed. Under increased reverse bias voltage at the enlarged depletion region width the EQE exceeded unity in the blue spectral region (300–550 nm) due to carrier multiplication and photoconductivity. The optical generation of additional electron–hole pairs within the depletion region still makes a significant impact onto the unbiased state. 

## Figures and Tables

**Figure 1 nanomaterials-08-01039-f001:**
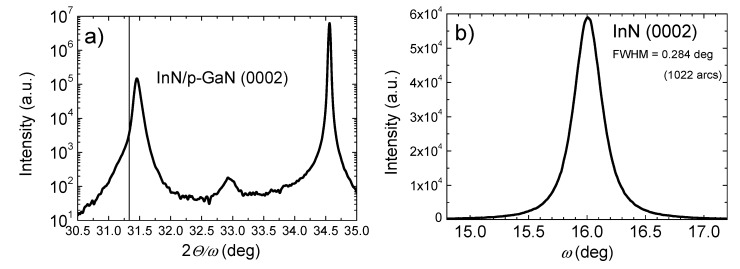
XRD measurement results of the InN/p-GaN structure at (0002) reflection: the 2-theta scan (**a**) and the rocking curve (**b**).

**Figure 2 nanomaterials-08-01039-f002:**
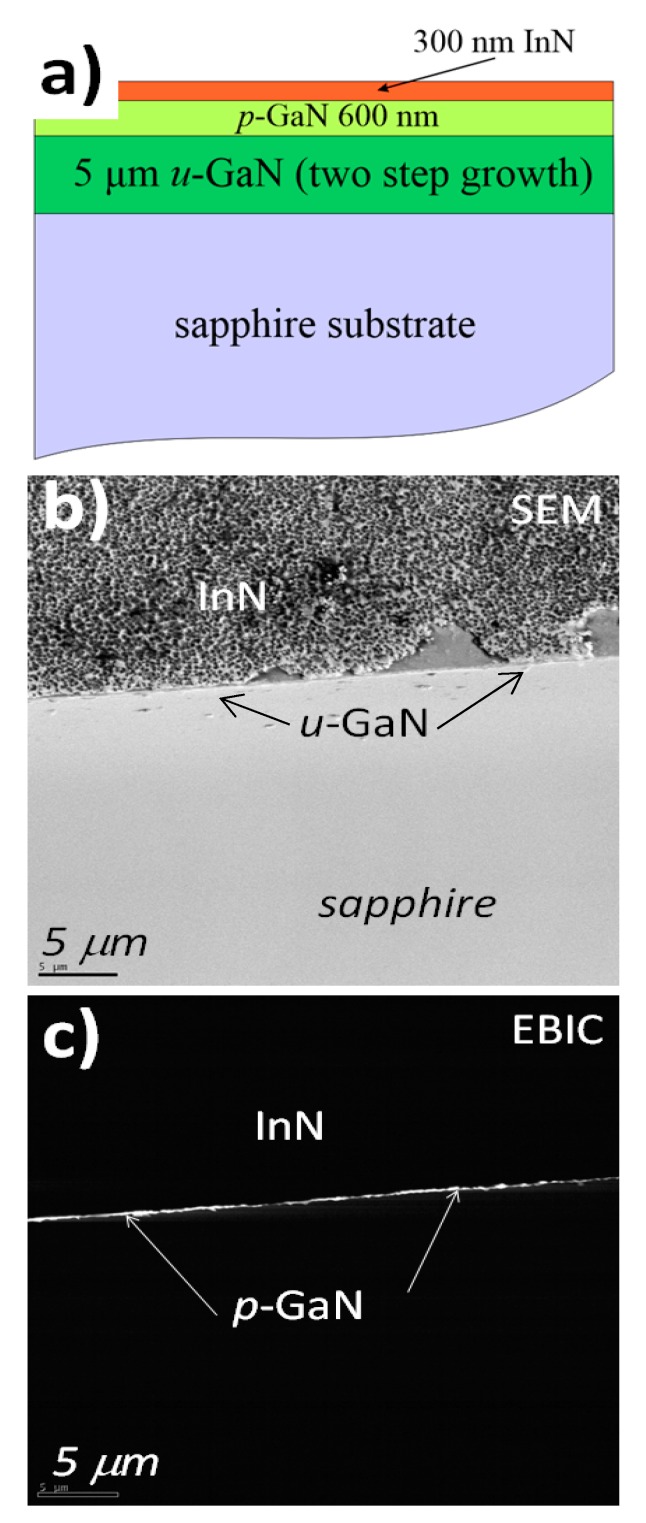
(**a**) Structure of the Mg doped p-GaN/InN junction; (**b**) SEM image (bird’s-eye view) of a cleaved p-GaN/InN heterojunction. (**c**) Electron-beam-induced current (EBIC) image superimposed on the SEM image. The bright line indicates the depleted region of the junction. Both SEM and EBIC images were obtained at 5 kV.

**Figure 3 nanomaterials-08-01039-f003:**
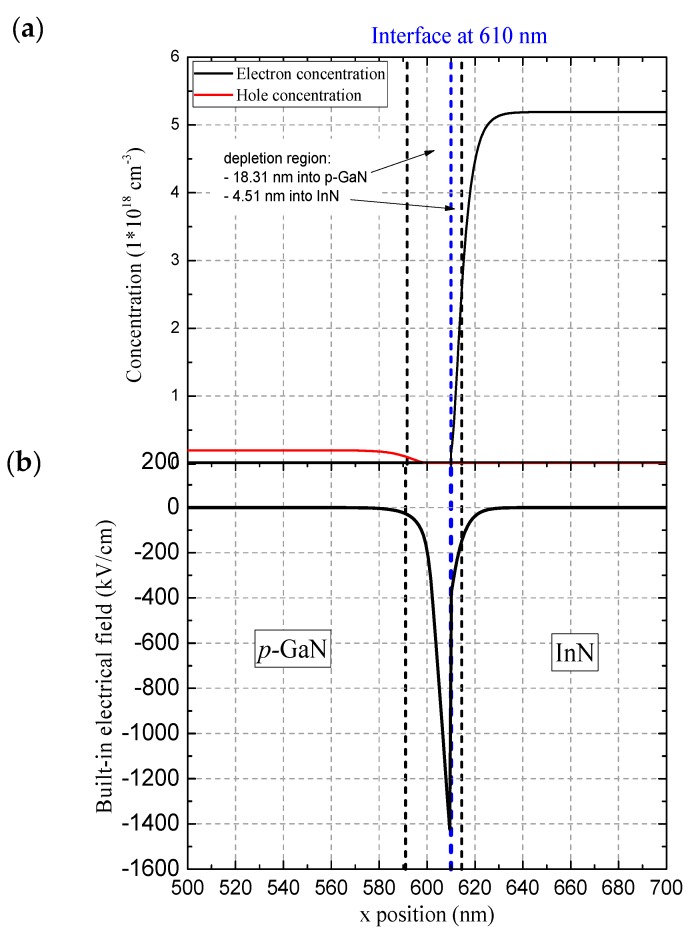
(**a**) Hole/electron concentration and (**b**) built-in electrical field as a function of depth at the InN/p-GaN interface.

**Figure 4 nanomaterials-08-01039-f004:**
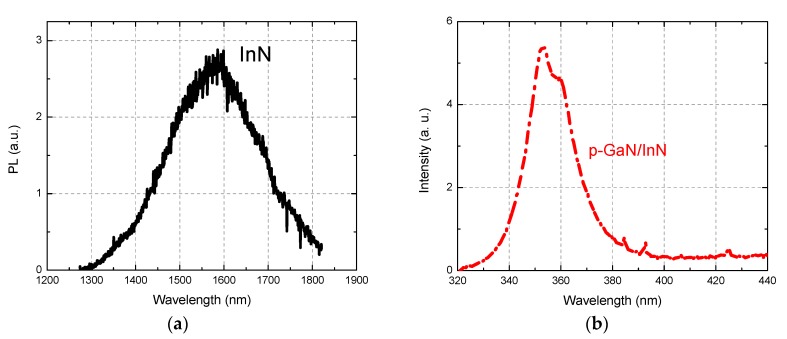
Room-temperature photoluminescence (PL) spectra of InN/p-GaN heterojunction grown on sapphire substrates: (**a**) near-infrared from 1200 nm to 1900 nm, (**b**) UV range from 320 nm to 440 nm.

**Figure 5 nanomaterials-08-01039-f005:**
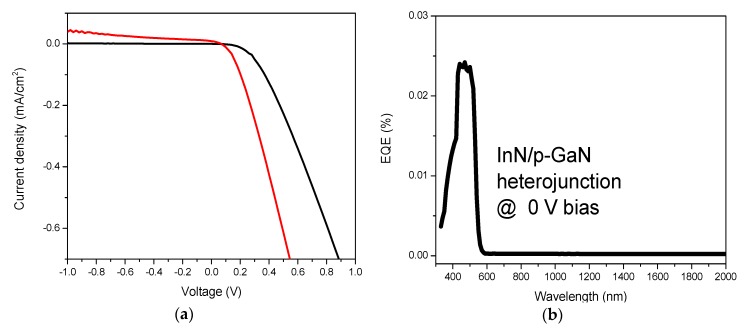
(**a**) Current voltage (J–V) characteristics of an InN/p-GaN heterojunction under dark (black) and AM1.5G illumination (red). (**b**) Corresponding external quantum efficiency (EQE) spectra of the unbiased InN/p-GaN heterojunction.

**Figure 6 nanomaterials-08-01039-f006:**
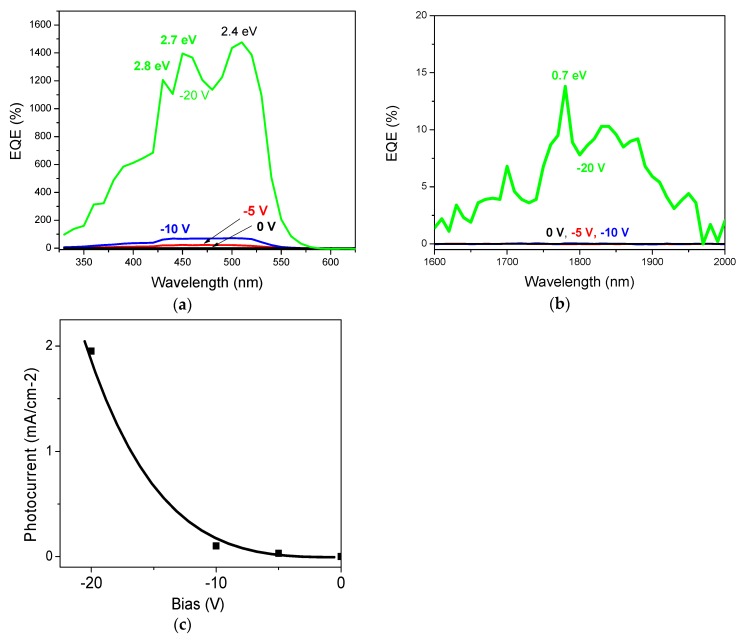
(**a**) EQE of the InN/p-GaN junction at the reverse bias of 0, −5, −10 and −20 V, (**b**) EQE spectra of the InN/p-GaN heterojunction at −20 V reverse bias in the NIR spectral region, (**c**) corresponding photocurrent density as a function of the reverse bias whereby the line is a guide to the eye.

**Figure 7 nanomaterials-08-01039-f007:**
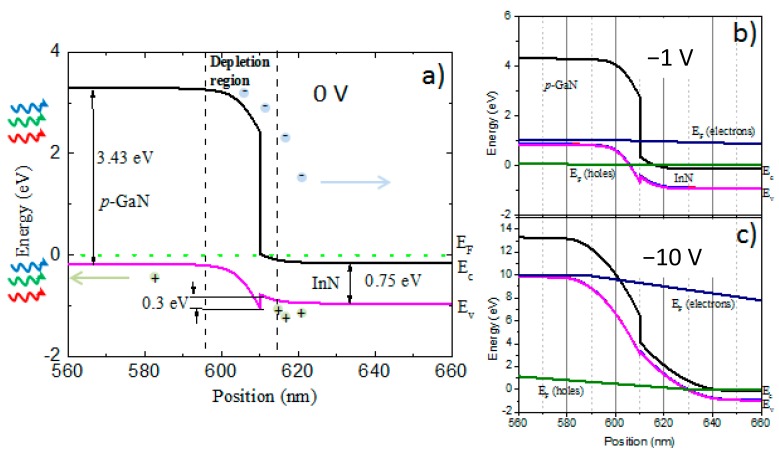
(**a**) Schematic of the unbiased (0 V) interface band alignment at room temperature for InN/p-GaN heterojunction. The irradiation of the junction from the GaN side is indicated. (**b**,**c**) Biased band alignments for the InN/p-GaN heterojunction at −1 V and −10 V reverse bias, respectively. The *E_C_* stands for conduction band, *E_V_* for valence band and *E_F_* for Fermi levels of electrons and holes.

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
