# Peer review of "Significant Carrier Extraction Enhancement at the Interface of an InN/p-GaN Heterojunction under Reverse Bias Voltage"

_nanomaterials, 2018, doi:10.3390/nano8121039_

Round 1
Reviewer 1 Report
The paper titled ‘Significant carrier extraction enhancement at the interface of an
InN/p-GaN heterojunction under reverse bias voltage’ was read thoroughly. The authors have studied the InN/p-GaN heterojunction interface grown using MOVPE. The study shows efficient charge extraction across interface along with enhanced rectifying behavior. Based on the overall quality of the paper and detail analysis of the interface, I would recommend the paper for publication.
Author Response
We thank the reviewers for the comments and valuable suggestions which have contributed to make our manuscript a stronger report. We have addressed all the queries and details of our changes and improvements are reported here below.
The paper titled ‘Significant carrier extraction enhancement at the interface of an
InN/p-GaN heterojunction under reverse bias voltage’ was read thoroughly. The authors have studied the InN/p-GaN heterojunction interface grown using MOVPE. The study shows efficient charge extraction across interface along with enhanced rectifying behavior. Based on the overall quality of the paper and detail analysis of the interface, I would recommend the paper for publication.
English language and style are fine/minor spell check required
The manuscript was checked by a native English speaking colleague.
Reviewer 2 Report
In this paper, Svrcek reports the growth of a superior quality InN/p-GaN interface using pulsed metalorganic vapor phase epitaxy (MOVPE). Using the system authors developed, good enhancement in the external quantum efficiency (EQE) in the blue spectral region (300-550 nm) was achieved. The paper is well written and presented very systematically. In particular, Introduction provides excellent information with very relevant reference citations. Considering the importance of the work presented in this paper, I would like to recommend acceptance for this paper. However before it can be published, a very minor revision is still required.
- The authors should re-plot all Figures in consistent format (font size and style). This will make this paper much easier to follow.
- Conclusion part of this paper can be improved by providing some better summary. Currently, the authors simply repeated what was written in the Abstract.
Author Response
We thank the reviewers for the comments and valuable suggestions which have contributed to make our manuscript a stronger report. We have addressed all the queries and details of our changes and improvements are reported here below.
Reviewer 2.
In this paper, Svrcek reports the growth of a superior quality InN/p-GaN interface using pulsed metalorganic vapor phase epitaxy (MOVPE). Using the system authors developed, good enhancement in the external quantum efficiency (EQE) in the blue spectral region (300-550 nm) was achieved. The paper is well written and presented very systematically. In particular, Introduction provides excellent information with very relevant reference citations. Considering the importance of the work presented in this paper, I would like to recommend acceptance for this paper. However before it can be published, a very minor revision is still required.
English language and style are fine/minor spell check required
The manuscript was checked by a native English speaking colleague.
- The authors should re-plot all Figures in consistent format (font size and style). This will make this paper much easier to follow.
Done it.
- Conclusion part of this paper can be improved by providing some better summary. Currently, the authors simply repeated what was written in the Abstract.
We corrected and improved the conclusion of the manuscript.
Reviewer 3 Report
The paper is well written and it does appear experimkentally sound. It is interesting that the nanamaterials generate EQE>100%. I recommend acceptance, provided the auhtros put more into context their photovoltaic techonlogy with other emerging cell technologies, such as bulk heterojunction organic cells, see for example:
doi: 10.1021/jacs.7b03412; 10.1021/acs.chemrev.8b00286
Author Response
We thank the reviewers for the comments and valuable suggestions which have contributed to make our manuscript a stronger report. We have addressed all the queries and details of our changes and improvements are reported here below.
Reviewer 3.
The paper is well written and it does appear experimkentally sound. It is interesting that the nanamaterials generate EQE>100%. I recommend acceptance, provided the auhtros put more into context their photovoltaic techonlogy with other emerging cell technologies, such as bulk heterojunction organic cells, see for example:
doi: 10.1021/jacs.7b03412; 10.1021/acs.chemrev.8b00286
We thanks reviewer for this suggestions, however we feel that the topic of the manuscript is not related to bulk heterojunction organic cells therefore we did not expand the context.
English language and style are fine/minor spell check required
The manuscript was checked by a native English speaking colleague.
Reviewer 4 Report
The manuscript present an interesting approch to obtain high quality InN epilayer based on the higher growth temperature as well as the pulsed growth mode.
Nevertheless, the following points should be considered to improve the manuscript:
-There is no information on the compositional sharpness of the GaN/InN junction; data should be presented about the possible formation of an interlayer at the junction of InGaN (and its thickness)
-Figure 2b should better indicate the InN film and the p-type GaN layer.
-The band gap of InN should be better identified; in the manuscript in different part is reported 1580 (which is about 0,78 eV) and 1780 (0.7 eV) and 0.75 in the band diagram in Fig.7
-The PL at 3.4 eV could be rather due to the undoped GaN, rather than to the p-GaN; better comments on the PL should be given to be related to the quality of the layers.
-The EQE efficiencies are peaked at the wavelength region of 400-500 nm; it would also be interesting to see if there is any PL in that range due to defects.
-The authors also state relates EQE blue spectra range peaks at 2.7-2.9 eV to compensating deep donors formed at high p-doping levels are responsible for the donor-to-band or donor acceptor-pair transitions; it would be interesting to see and correlate to PL spectra in that region.
Author Response
We thank the reviewers for the comments and valuable suggestions which have contributed to make our manuscript a stronger report. We have addressed all the queries and details of our changes and improvements are reported here below.
Reviewer 4.
The manuscript present an interesting approch to obtain high quality InN epilayer based on the higher growth temperature as well as the pulsed growth mode.
Nevertheless, the following points should be considered to improve the manuscript:
-There is no information on the compositional sharpness of the GaN/InN junction; data should be presented about the possible formation of an interlayer at the junction of InGaN (and its thickness)
The issue related to GaN contamination by the In during InGaN/GaN MQWs growth is investigated by other groups and can be found in the literature. The interface has a sharp transition when InGaN is grown on GaN (for example see [Galtrey JAP]). Based on studies we think that the interface of InN grown on p-GaN is abrupt. The abruptness can be examined using TEM. However, this interlayer consisting of several atom planes, if any, presents only a very small part of the depletion region, therefore negligible impact on the characteristics and was not included in the model.
Galtrey, M. J., R. A. Oliver, M. J. Kappers, C. J. Humphreys, P. H. Clifton, D. Larson, D. W. Saxey, and A. Cerezo. Three-dimensional atom probe analysis of green and blue-emitting InxGa1−xN∕ GaN multiple quantum well structures. Journal of Applied Physics, 104(1), 013524 (2008). https://doi.org/10.1063/1.2938081
-Figure 2b should better indicate the InN film and the p-type GaN layer.
The u-GaN and p-GaN are not possible to discern by SEM due to z-contrast issues, therefore we only approximately indicate the p-type GaN region by arrows, taking into account it’s 600 nm thickness.
-The band gap of InN should be better identified; in the manuscript in different part is reported 1580 (which is about 0,78 eV) and 1780 (0.7 eV) and 0.75 in the band diagram in Fig.7
The PL measurement and the EQE measurement have different charge carriers injection mechanisms, the first exciting them optically, the second – electrically. Under applied voltage, the EQE signal corresponds just to the right-wing side of the broad PL spectra peaked at 0.78 eV. Therefore, in modeling we’ve chosen some average value 0.75 eV.
-The PL at 3.4 eV could be rather due to the undoped GaN, rather than to the p-GaN; better comments on the PL should be given to be related to the quality of the layers.
Typically, the p-GaN bandgap related PL spectra have much broader tail towards lower energies remaining the same bandgap peak as for undoped GaN. Our rather steep tail represents relative good quality of the p-GaN.
-The EQE efficiencies are peaked at the wavelength region of 400-500 nm; it would also be interesting to see if there is any PL in that range due to defects.
We include new Fig. 5 b that show no substantial EQE till 2000 nm i.e. also no PL.
-The authors also state relates EQE blue spectra range peaks at 2.7-2.9 eV to compensating deep donors formed at high p-doping levels are responsible for the donor-to-band or donor acceptor-pair transitions; it would be interesting to see and correlate to PL spectra in that region.
No substantial PL in this spectral region was observed (p-GaN is of good quality) as seen in Figure 5b. It features similar behavior as observed for InN, where the electrically driven EQE shows redshifted spectra in respect to the optically PL.
English language and style are fine/minor spell check required
The manuscript was checked by a native English speaking colleague.
Round 2
Reviewer 2 Report
The paper can now be accepted for publication.
Reviewer 3 Report
OK
Reviewer 4 Report
The authors have clarified all issues in the revised version as well as their response.
Therefore the manuscript is now suitable for publication